# YouTube Channels, Subscribers, Uploads and Views: A Multidimensional Analysis of the First 1700 Channels from July 2022

**Dana Adriana Lupșa-Tătaru and Radu Lixăndroiu ***

Departament of Management and Economic Informatics, Transilvania University of Brasov,
500036 Brasov, Romania
* Correspondence: lixi.radu@unitbv.ro

**Abstract:** In a world of online social life and social media development, more people are interested in developing video content-based businesses, including YouTube channels, and information sharing with the main purpose of making money. This paper conducts a multidimensional analysis of the first 100 YouTube channels (July, 2022) from each of the 17 domains identified, using SocialBalde.com. The purpose of the paper is to investigate the crucial factors to take into consideration when starting a social media business on YouTube, namely the domain and description of the channel. The objective is to equip future social media entrepreneurs with two elements that should be considered when starting an online business. We perform a two-fold analysis, by exploring correlations between data for each channel and conducting a semantical analysis of the text describing each channel. In spite of the numerous research papers related to YouTube, none of them focus on this topic, i.e., the practical transformation of information into data that can be measured with multidimensional instruments. The results have interesting implications regarding how a successful channel should be developed; we also present sustainable guidelines for social media entrepreneurs to follow during the process of starting a business. Consequently, a successful online business is partly the result of the domain chosen and the channel description.

**Keywords:** YouTube; social media; entrepreneurship

## 1. Introduction

The development of social media and communication based on technology using the Internet has given rise to a new generation of entrepreneurs, also known as "social media entrepreneurship". As technology is used in order to create and exchange extremely easy content generated by users [1], businesses have viewed social media as a means of connection with their audiences and thus use it for marketing, communication, innovation and networking [2–6].

The new generation of entrepreneurs has recently become an area of interest, since the model of their businesses is based on their social media network and the way they convert their social media activity into business opportunities. Social media is an environment in which consumers are empowered to create content; thus, they may become influencers—people with the power to shape other consumers' attitudes and behaviors [7,8]. Because they possess this influence, they become drivers for marketing other brands, and thus are financially rewarded [8,9].

Most of them are paid to use their social network for targeted advertisements or sponsored content. Some are launching their own brand's products or services, targeted directly at their audiences, and their personal brands become impactful through content creation.

Businesses select these people based on their popularity and, relatedly, based on their engagement scores and statistics, which are reflected in their audience's reactions—likes, love, comments, shares and retweets [7,10]. The more engaged a person is, the more

popular [11,12] they are and the more likely they are to have active followers [10,13]; thus, influencers can connect the promoted brand with consumers [9,14] and influence consumers' opinions and intentions to buy.

A very special feature and also the foundation of social media entrepreneurship according to [1], p. 70 is the use of the emotional connections between these entrepreneurs and their followers in order to capitalize on these connections [15,16].

YouTube is a dominant and very large social media platform; it is different to Facebook and Twitter in terms of interest from researchers over the years. Interest has increased recently and now includes themes such as political content [17] or deception [18], the implications of the algorithm in terms of politics and culture [19,20], and the controversies regarding social media's roles in radicalization, misconduct and abuse. Additionally, qualitative research has been conducted that examines the daily use of YouTube by professionals and nonprofessionals [21–24] as well as the importance of daily use for entertainment, politics and the economy [25–29].

At present, the theoretical and empirical literature on the economics of social media entrepreneurs has addressed this phenomenon to a limited extent. This paper, therefore, aims to extend our understanding of the phenomenon of social media entrepreneurs. Our research questions refer to the description accuracy of YouTube channels and to their profitability in terms of social media entrepreneurship. In order to investigate the themes of the research, the paper provides a mathematical analysis of the first 1700 channels according to SocialBlade (June 2022) from 17 different video categories.

Even if YouTube success has been examined in the specialized literature [30], not enough is known about how a channel can gain value and which special category social media entrepreneurs should choose to grow their channel in order to be successful—in terms of views, subscribers and uploads. Some studies have identified variables that have to be addressed when creating scientific video content [30], while other studies have explored the characteristics of scientific channels' video content as means to improve engagement [31] and the type of content [32]. Some have explored the factors influencing the relations between celebrities—very famous personal brands—and users' intention to purchase on social network services [33,34], while others have concluded that entertainment and informativeness represent the key factors of sponsored content value that affect attitudes towards social/personal brands on YouTube [35].

Our theoretical contributions consist of covering the gap in knowledge between how to choose a domain for a YouTube channel and how to write a description when starting a business. It should be noted that there has been no previous research on this topic using the methodology described and the instruments used in this research. The input we used was transformed into data that were statistically analyzed through a semantical analysis, and the data regarding subscribers, views and uploads were measured via the TOPSIS method.

## 2. Theoretical Background

### 2.1. Social Media Entrepreneurship

The modern economic and entrepreneurial theory of social media entrepreneurs is based on the development of social media platforms and the theoretical work of [36–38], which outlines the link between the emergence of these platforms and this division of digital media entrepreneurship. As a definition, digital media entrepreneurship emphasizes creating and selling new digital platforms and other digital products, or creating value through the use of existing digital platforms [39,40]. This paper focuses on existing social media platforms, namely YouTube. The growing importance of social media platforms for the development of social media entrepreneurship has been a very interesting topic for researchers lately [41–43]. As social media entrepreneurs move toward commercialization, the success and influence of social media entrepreneurs are related to the number of followers and the number of reactions (likes, love, shares, tweets, retweets and comments) [44].

Besides Facebook and Twitter, YouTube is a video-sharing social media platform that became a dominant platform in terms of hosting millions of channels, billions of videos and more than two billion active users every month. The empirical research conducted by now shows that there is some interest in the social network structure [45] and in the content analysis of the most popular videos [27]. As a platform, YouTube has evolved from video-sharing to the monetization of channels, thus generating, in 2019, USD 15 billion from advertising, representing almost 10% of Google's overall revenue. This transformation through profitability leads to design, content and audience changes and thus to strict rules regarding advertising and the 'professionalization' of YouTube's content creators. Thus, nowadays, the concept of social media entertainment [46] is used when presenting popular types of content on YouTube. Even though the YouTube video content is, to some extent, professional, it is still produced by amateurs, by star YouTubers starts, television networks or music producers, to reach large audiences, especially younger viewers.

### 2.2. Factors Influencing Social Media Entrepreneurship

The connection between the development of social media entrepreneurship and the followers of social media entrepreneurs was previously centered on the importance of authenticity [21,47], which is a factor that indicates whether a person is indeed what he or she claims to be [48,49]. Authenticity may be described as "being true to himself/herself", "being real", "actioning based on feelings" [48]. The author presented the way in which social actors, including social entrepreneurs, create a representative public image for their private identity with a believable and approved role by the followers. [49,50]. This means that authenticity is connected to follower's perceptions of being real and sincere.

In order to be perceived as authentic, the social entrepreneurs' strategy is to present "intimate details of their thoughts, dreams, food consumption, and they present personas that appear to be less controlled than those of highly regulated, highly consumer brand-oriented film and television celebrities" [51], p.346. As studies have shown, if the followers are consuming these aspects in a repetitive way, they feel as if they are closely connected to social media entrepreneurs [51].

Social media entrepreneurship is challenging, since the commercial goals influence authenticity [51]. Thus, if the social entrepreneur is perceived as inauthentic, their connection with followers is affected and also the business, according to self-determination theory [52,53]. In order to overcome this challenge, social media entrepreneurs have to obtain the followers' acceptance and legitimization of commercial objectives and monetization [15].

YouTube is an important platform used by brands to promote their products and services through the influence of people's large number of followers and the opinion's influence of acquisition process [54,55]. YouTube is popular because the content created by these people seems to be more realistic than traditional advertising [56] and because the engagement is encouraged through commenting and social interactions [57]; therefore, this research is limited to YouTube channels only.

Since 2005, the year it was launched, YouTube has evolved from a place where amateurs post ad-free videos to a platform populated by commercial and professional videos. It is a place where an individual can develop a business from a personal brand through video content generation and monetize their influence with paid ads or their own branded products or services [58], p. 114, [59], p. 56. This opportunity of monetizing content is also a trend, and it enabled personal brand channels to grow into the source of content creators' income [60], p. 378.

Given the critical factor of a mass audiences' views, and the importance and attention given to these videos [61–63], YouTube is a state-of-the-art instrument and provides the opportunity for any creator to develop a personal brand and become a monetized social media entrepreneur [64,65].

## 3. Methodology

Because of the literature gap regarding the important factors needed to become a successful start-up social media entrepreneur, we formulated two hypotheses regarding the following two interconnected factors: the domain of the channel, and the description of the channel. The hypotheses of the research are as follows:

**H1.** *The more accurately described a YouTube channel is, the more successful it is.*

**H2.** *The most successful channels are the ones from the following categories: entertaining, travel and personal vlogs. Success is defined, in this case, as the number of views, subscribers and uploads.*

The first stage of the analysis consists of choosing 1700 YouTube channels. We selected 100 out of each of the 17 different categories based on their SB (Social Blade) ranking. The ranking was calculated by SB based on the number of videos uploaded, the number of subscribers and the number of views. The categories were as follows: auto and vehicles, comedy, education, entertainment, film, gaming, how to and style, made for kids, music, news and politics, nonprofit and activism, people and blogs, pets and animals, science and technology, shows, sports and travel.

The reason for using Social Blade as an instrument for this research is that SB is a new and modern instrument mostly used by YouTube start-up social media entrepreneurs; the American platform was founded in 2008 and it is widely recognized for its usefulness in statistical studies. It also offers information about subscribers and includes information such as estimated earnings and future projections, providing both numerical data and easy-to-read graphs. SB was used to analyze data from 1700 YouTube channels. SB, certificate YT, compiles data from YouTube, Twitter, Twitch, Daily Motion, Mixer, and Instagram and uses the data for statistical graphs and charts that track progress and growth. Social Blade offers statistics and is currently tracking 23+ million YouTube channels, 6+ million Twitter Profiles, 5+ million Twitch channels, 206+ thousand Daily Motion users, and 416+ thousand Mixer Streamers.

The second stage of the analysis was conducted with the following two steps: (a) multidimensional data analysis using Tableau Public for all of the 1700 channels and (b) channel description text semantic analysis, using Meaning Cloud, for 170 selected channels.

The third stage of the analysis was to use the TOPSIS method to calculate the distance from each channel's data to an ideal positive and an ideal negative channel's data. Using the TOPSIS method to compare the channels with an ideal represents an adequate instrument to study their hierarchy. It is a method of compensatory aggregation that compares a set of alternatives by identifying the weights for each criterion, normalizing scores for each criterion and calculating the geometric distance between each alternative and the ideal alternative, which is the best score in each criterion. The instruments—multidimensional data analysis and semantical analysis—are modern and extremely efficient for analyzing large data volumes.

The purpose was to identify the combination of factors that directly determine the success of a channel, defined, in this case, as number of views, subscribers and uploads, all leading to profit.

### 3.1. The Multidimensional Data Analysis

For the multidimensional data analysis, the data retrieved from SB were converted into an OLAP data cube, and then Tableau Public was used. For each of the 1700 channels, we considered the number of videos uploaded, the number of subscribers and the number of views. These numerical dimensions were used to conduct a statistical analysis; more precisely, we determined the relative correlations between the above mentioned dimensions.

Multidimensional data analysis refers to analyzing and categorizing data, based on multiple factors related to various entities, called dimensions. This method of analysis is an instrument used for different processes of mathematical modelling, with large and complex data. In this particular case, it provides insights across the gamut of YouTube channels.

*3.2. The Text Semantic Analysis*

From the total of 1700 channels, 170 were selected using a pace of 10 units and considering only the English described channels. We elapsed a unit if another language description was met because the semantical analysis tool, called Meaning Cloud (MC), only allows for the analysis of a few language descriptions, including English, French, Italian and Spanish.

A YouTube channel description is similar to the About Page of a website and explains to potential viewers what the content is about, including the issues tackled and the communities served.

YouTube descriptions should be a primary interest in YouTube marketing strategies, because video descriptions are important both to SEO keywords search results and to encourage viewers to spend more time on the video.

Meaning Cloud (MC) is an instrument that uses deep categorization analysis which integrates the functionality provided by the Deep Categorization API. It assigns one or more categories to a text using a detailed rule-based language that allows for very specific scenarios and patterns identification, with a combination of morphological, semantic and text rules.

Deep Categorization includes the Code (shows the code associated to the category), Label (shows the label of the category; the label is non-configurable, so it always appears in the results), Rank (shows the rank or order in which a category has been associated to a text), Relevance (shows the absolute relevance associated with the category), Relative Relevance (shows the relative relevance associated with the category) and Polarity (shows the polarity of the category detected).

The data obtained using Meaning Cloud, and consequently Deep Categorization, were analyzed for each channel by using the TOPSIS method, in order to calculate the distance of each channel's data from a positive and a negative ideal channel's data.

The TOPSIS method—the Technique for Order Preference by Similarity to Ideal Solution—is based on the research of [52] which considers that the ideal positive and negative solution are determined from the values of different criteria options. The options are ranked based on the distances between those two solutions—ideal positive and ideal negative ones. In order to conduct a comparative analysis, a relative distance between each solution and the positive ideal solution is calculated. Geometrically, each option is a n-dimensional space point, where n is the number of criteria. In this space, two other points are defined (positive ideal solution and negative ideal solution), related to which the relative distances of the options are determined.

The algorithm of the method consists of the following steps. The normalized matrix is obtained, using vector normalization, according to $R = (r_{ij})$, $i = 1, 2, \ldots, m$, $j = 1, 2, \ldots, n$. The normalized weighted matrix is obtained according to $V = (v_{ij})$, $i = 1, 2, \ldots, m$, $j = 1, 2, \ldots, n$, where $v_{ij} = p_j \cdot r_{ij}$ and $P = (p_1, p_2, \ldots, p_n)$ are the vectors of the importance coefficients (objective, subjective or aggregated). Next, the ideal positive solution (Vid) and the ideal negative solution (Vne) are determined, where $V^{id} = \left( V_1^{id}, V_2^{id}, \ldots, V_n^{id} \right)$, $V^{ne} = \left( V_1^{ne}, V_2^{ne}, \ldots, V_n^{ne} \right)$ and $V_j^{id} = \max_{1 \leq i \leq m} V_{ij}$, and $V_j^{ne} = \min_{1 \leq i \leq m} V_{ij}$. The distance between the options and the ideal negative solution can be determined as follows: $d_i^{id} = \sqrt{\sum_{j=1}^{n} \left( V_{ij} - V_j^{id} \right)^2}$ and $d_i^{ne} = \sqrt{\sum_{j=1}^{n} \left( V_{ij} - V_j^{ne} \right)^2}$, $i = 1, 2, \ldots, m$. Then, the relative proximity to the ideal solution is calculated, according to $e_i^{id} = 1 - \frac{d_i^{id}}{d_i^{id} + d_i^{ne}} = \frac{d_i^{ne}}{d_i^{id} + d_i^{ne}}$, $i = 1, 2, \ldots, m$, resulting in $0 s e_i^{id} \leq 1$.

Eventually, the *V* set of options is determined according to the descending values calculated in the previous step.

For the TOPSIS analysis, the label, rank, relevance, relative relevance and polarity items from MC were used, along with the Channel Type from SB. Thus, the results showed the extent to which the identified category corresponds to the category itself from SB, using MC for the description of each channel.

## 4. Results

*The Results of The Multidimensional Data Analysis*

Using a Tableau Public and comparative analysis of the SB ranking for different categories of YouTube channels, we obtained the initial results. The results show that there is a great difference between educational channels, mostly ranked A or A+ and nonprofit or activism channels, ranked B+ (Figure 1).

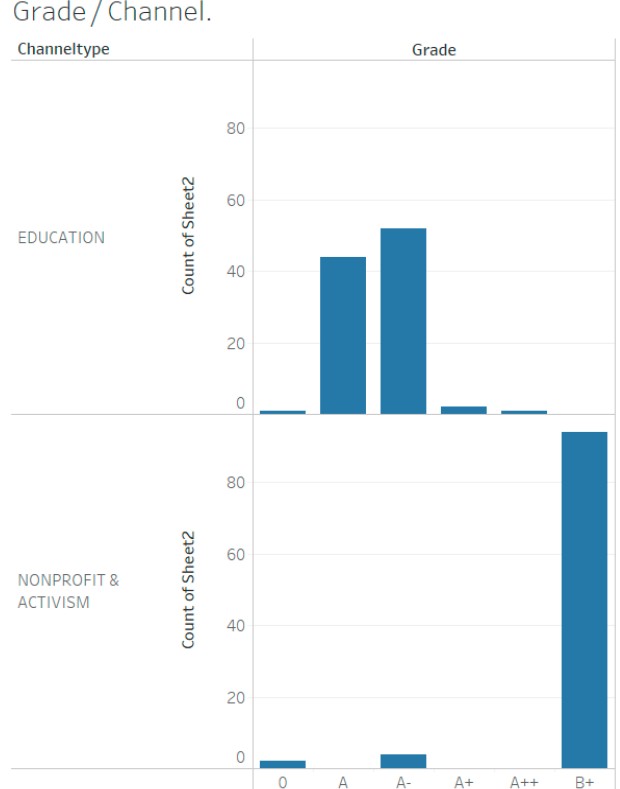

**Figure 1.** Comparative analysis of ranking different YouTube categories of channels.

The following image (Figures 2–4) represents the analysis of all 17 YouTube channel categories from the point of view of ranking.

By analyzing the number of videos uploaded to each channel, we obtained further results. Thus, the first place is occupied by the news category, which is in agreement with the results shown in Figure 4, even though, from the point of view of the number of views, the first place is not occupied by the same category. The explanation is that most people want to be informed and when they think that they are informed, they do not play the video again, in contrast to the repeated views of music, entertainment, movies, and educational videos.

By calculating correlations between existing data, other results were obtained. Thus, there is an obvious, strong correlation between the number of subscribers and the number of views.

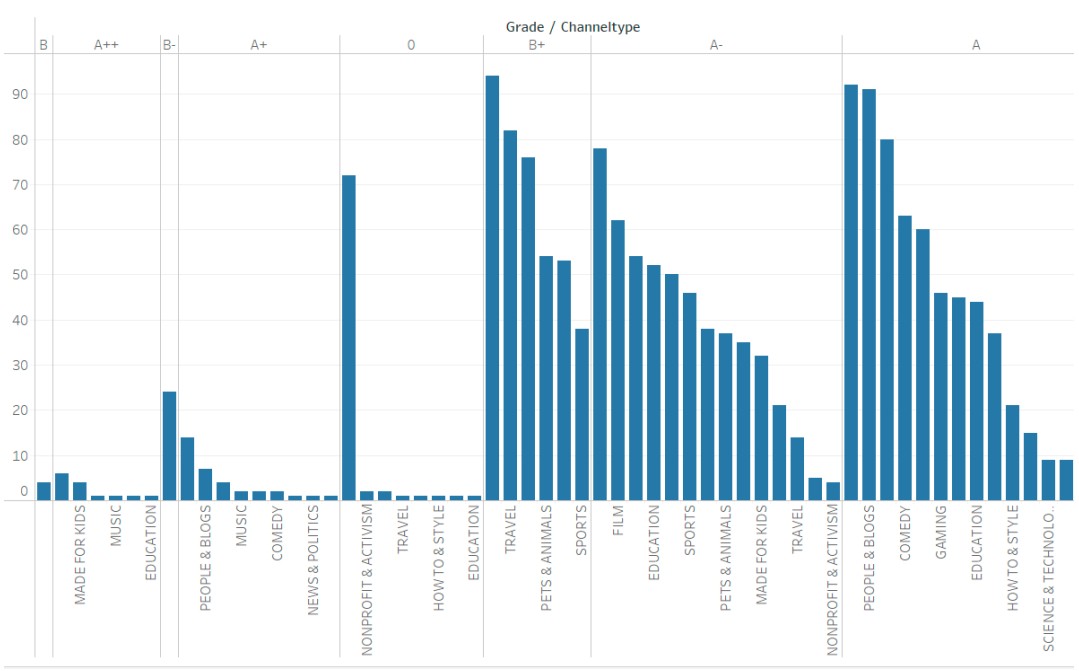

**Figure 2.** The distribution of the ranking for all the 17 categories of YouTube channels.

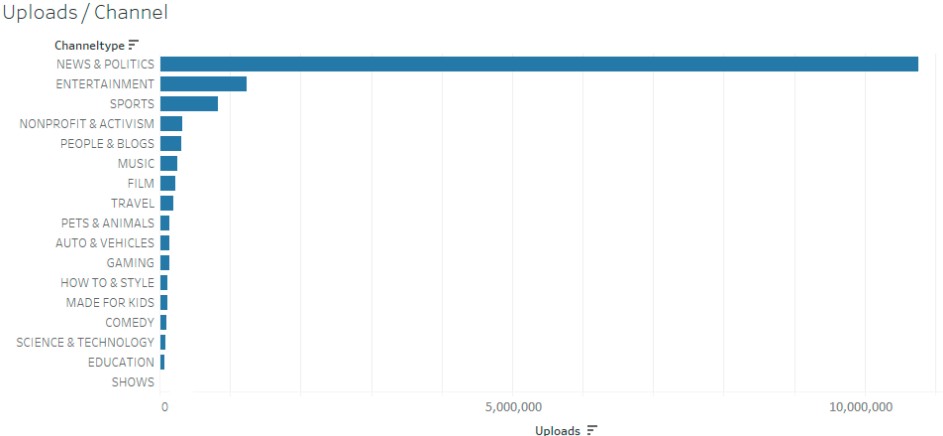

**Figure 3.** The number of videos uploaded on each channel.

There is also a weak correlation between the number of videos uploaded and the number of views, which leads to the conclusion that subscribers are loyal to channels when watching their videos; however, when the content is no longer interesting to them, they unsubscribe.

There are also channels with more uploaded videos but less subscribers and less views; this signifies subscribers' lack of interest.

There is a strong correlation (Table 1) between the number of subscribers and the views (0.920) and almost no correlation between the number of uploads and the number of views (0.095). Data are real, relevant and very up to date; thus, the results may be used by start-up social media entrepreneurs when making decisions for new sustainable managerial behaviors.

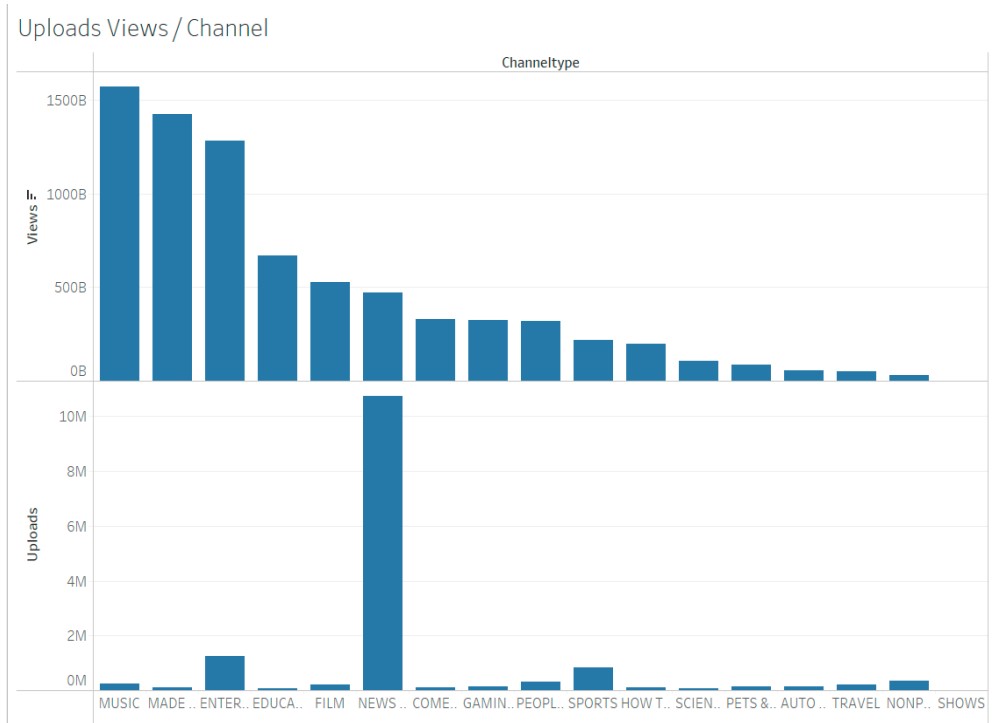

**Figure 4.** The number of videos uploaded vs the number of views.

**Table 1.** The results of the semantic analysis of the text.

| Correlation | Value |
|:---:|:---:|
| Corr Rank—Subs. | −0.191 |
| Corr Rank—Uploads | −0.074 |
| Corr Rank—Views | −0.200 |
| Corr Subs—Views | 0.920 |
| Corr Uploads—Views | 0.095 |

Two solutions (Table 2) were generated as a result of using a TOPSIS analysis—the positive ideal and the negative ideal; we calculated the distances between these two solutions and each of the 170 analyzed channels.

**Table 2.** The TOPSIS analysis.

| Username | Relevance | Relative Relevance | Polarity | Polarity | Type vs. Code |
|:---:|:---:|:---:|:---:|:---:|:---:|
| **IDEAL+** | 1 | 100 | 5 | 5 | 1 |
| **IDEAL−** | 10 | 0 | | 1 | 0 |

Where Channel Type vs. Label is measured as 1 or 0 after comparing the results obtained for Label to the ones of Channel Type from SB, where 1 means perfect identification and 0 means wrong identification.

In order to obtain the polarity level, the text codification was transformed into numbers from 1 to 5, where 1 is a very negative text and 5 is a very positive text (Table 3).

**Table 3.** Top 10 YouTube channels in terms of the distance between the ideal solution based on the description.

| Grade | Username | Label | Rank | Relevance | Relative Relevance | Polarity | Polarity Level | Type vs. Code | $d_i^{id}$ | $d_i^{ne}$ | $e_i^{id}$ |
|---|---|---|---|---|---|---|---|---|---|---|---|
| A | Goldmines Bollywood | Events and Attractions > Cinemas and Events | 1 | 1 | 100 | P+ | 5 | 1 | 0 | 100.489 | 1 |
| B+ | Monkey Magic | Movies > Action and Adventure Movies | 1 | 1 | 100 | P+ | 5 | 1 | 0 | 100.489 | 1 |
| A | Elsa Arca | News and Politics > Crime | 1 | 1 | 100 | P+ | 5 | 0 | 1 | 100.484 | 0.99015 |
| B+ | Jesus Image | News and Politics > Politics | 1 | 1 | 100 | P+ | 5 | 0 | 1 | 100.484 | 0.99015 |
| A | News24 | Business and Finance | 1 | 1 | 100 | P+ | 5 | 0 | 1 | 100.484 | 0.99015 |
| A | SOMOY TV | Hobbies and Interests > Collecting | 1 | 1 | 100 | P+ | 5 | 0 | 1 | 100.484 | 0.99015 |
| A+ | Zee Kids | Travel > Travel Type | 1 | 1 | 100 | P+ | 5 | 0 | 1 | 100.484 | 0.99015 |
| B+ | AutoTopNL | Automotive > Auto Body Styles | 1 | 1 | 100 | P | 4 | 1 | 1 | 100.454 | 0.99014 |

Based on the results, the conclusion is that the usage of key words to describe each YouTube channels category is essential. As observed in Table 1, the first two positions are occupied by the channels that have ideal values, meaning that the description is representative for the channel; thus, they are allocated to the positive category, as they send a very positive message and have 100% relevance.

In contrast, the results show YouTube channels that have no description have a maximum distance from the positive ideal.

## 5. Conclusions

According to the YouTube Partner program, once a channel has 1000 subscribers and more than 4000 valid public watch hours in the most recent 12 month period, it becomes eligible for greater access to YouTube resources and monetization features. Therefore, it becomes successful, according to the definition of the above research.

In order to achieve 1000 subscribers and more than 4000 valid public watch hours, the channel has to have certain features, such as the following: attractive thumbnails—the way the results of searches appear for people who decide to view a video—and a perfect pictogram for the channel—the pictogram provides a logo for branding. Another important element is a list of videos. This is the best way to keep people engaged when watching videos on a channel and a good way to minimize the chances that they leave to watch another channel's videos. In order to engage people when they watch a YouTube channel for the first time and to make them curious about the content, it is recommended to create a trailer—which is similar to hotshots for movies. Creating constantly interesting content, with an attractive presentation, excellent branding, appropriate music and clear sound, is another way obtain more subscribers. Scheduling the videos in order to attract more views is another great way to create a successful YouTube channel, along with defining the target market by both asking what kind of content the audience wants to view and using YouTube analytics. In order to organically obtain more views, creators have to grow their list of subscribers; a simple way to do this is to ask the viewers to subscribe.

In order to establish a profitable business, a social media entrepreneur needs an entertainment, travel or personal YouTube channel on which videos are constantly uploaded; their content should be perceived as engrossing by their subscribers. A profitable social media entrepreneur, in this case, is defined as somebody who is paid to advertise products or services or launches their own products or services. To obtain a large number of subscribers and views, interesting videos should be uploaded consistently and also with a proper description of the channel so that content can easily be found in the right category.

This research bridges the existing gap in the literature by studying the importance of choosing the domain for a YouTube channel and writing a description when starting a social media business. Previous research did not touch on this topic using the methodology described and the instruments employed in this research. Even so, the paper is only a starting point, as the analysis will be continued for other social media channels as well.

The hypotheses are confirmed by the research. Thus, the more accurately described a YouTube channel is, the more looked up it is; furthermore, the most successful channels, in terms of number of views, subscribers and upload are the ones in the following categories: entertainment, travelling and personal vlogs. The research represents a starting point for any social media entrepreneur wanting to start a YouTube channel as a business. Practical implications consist of the insight that social media entrepreneurs have to pay attention to the accuracy of the channel description and should choose the proper domain to start an efficient and sustainable business. Future research will be conducted, since all social media channels provide innovative business opportunities.

Theoretical implications concern the development of social media entrepreneurship theory; one may conclude that starting a business on social media channels, such as YouTube, assisted by entrepreneurship strategies already known and enshrined (Drucker, 1999) is a new entrepreneurship strategy. As for methodological implications, social media entrepreneurs should pay attention to the description of the channel from an online-business start-up point of view.

## 6. Discussion

As for the limits of the study, the first would be that as educational channels and nonprofit or activism channels are managed by professionals, the videos are more structured and the concepts are very clearly explained. The factors related to the great difference of ranking can only be anticipated. The analysis of these factors, thus, will be the focus for further research papers, as distinct research will be conducted in the future.

In future studies, the results obtained within this paper will be enriched with the use of Kendall's coefficient of concordance.

The study was conducted by only using YouTube as a source for data; if we had used different sources, the results would have been different. Additionally, the research above is centered on English YouTube channel descriptions, and if we had used other descriptions, on other languages, the ensemble images would have been more notable.

**Author Contributions:** Conceptualization, D.A.L.-T. and R.L.; methodology, R.L.; formal analysis, R.L.; investigation, R.L.; resources, D.A.L.-T.; data curation, D.A.L.-T.; writing—original draft preparation, D.A.L.-T. All authors have read and agreed to the published version of the manuscript.

**Funding:** The research was financed by Transylvania University of Brasov, Romania.

**Institutional Review Board Statement:** Not applicable.

**Informed Consent Statement:** Not applicable.

**Data Availability Statement:** Not applicable.

**Conflicts of Interest:** The authors declare no conflict of interest.

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
