# Peer review of "YouTube Channels, Subscribers, Uploads and Views: A Multidimensional Analysis of the First 1700 Channels from July 2022"

_sustainability, doi:10.3390/su142013112_

Round 1

Author Response

Dear Reviewer 1,

Thank you very much for your reviews! Your observations are very valuable for us, we think that they enriched our research and that the insights and the results will be even better received, understood and even more useful for scientist and large public as well, thus, we have been reading and solving each one, very carefully. 

In the following part, we tried to answer your requirements.

This paper analyses the top 100 YouTube channels from each of 17 domains using SocialBlade, looking for correlations between data for each channel and analyzing the descriptive text for each channel.

The suitability for this journal is unaddressed.

We have added on the abstract explanations regarding the sustainability feature of an online start-up, since, because of the findings of the research, the new social media entrepreneurs on YT will know what domain to choose and how to describe the channel in order to have success.

For instance, we have described here: “presenting sustainable guiding lines for social media entrepreneurs to follow on the demarch of starting a business, since they know which domain to choose and how to describe the channel, in order to have a successful online business.

The topic could be of interest to readers. YouTube is a relevant and worthwhile topic.

The changes suggested have been done, thus, the cursivity and comprehensibility of the text were improved.

There are considerable problems with the writing style of the paper. It is consistently very difficult to follow the meaning. Typos always exist, but there are numerous spelling and grammatical errors in the paper, beginning the Abstract with the spelling of the analytical tool.

The style of the paper was modified; thus, the comprehensibility and the coherence of the text were improved.

The structure of the paper is IMRAD structure, but there is a notable absence of content reviewing the existing literature to build a foundation for Hypothesis development, identification of the gap, and justification for methodology, or the study in general.

We have added Content about literature review in order to build a foundation for hypothesis development. The strength of the paper and the need of this study are that there is no other research on this particular topic – how to choose a domain and how to describe the channel when starting a YT business as a social media entrepreneur.

The explanation for the using of SB, as instrument for the research, was added - The reason for using Social Blade as instrument for this research is that SB is an American platform founded in 2008, widely recognized for statistical studies, a new and modern instrument used mostly by social media entrepreneurs that start a new business on YouTube. It offers information about subscribers, too.

As for the methodology, we added explanations – we consider that using Topsis method for comparing the channels with an ideal, is a very proper instrument to study their hierarchy. It is a method of compensatory aggregation that compares a set of alternatives by identifying weights for each criterion, normalizing scores for each criterion and calculating the geometric distance between each alternative and the ideal alternative, which is the best score in each criterion. 

The Theoretical contribution is apparent in neither the Introduction, nor the Discussion. Section 2 seems to be a continuation of the Introduction, describing social media entrepreneurship, and FB with no mention of theoretical foundation. Authenticity is mentioned, but is not revisited in the manuscript.

We have added explanations regarding theoretical contributions, both on Introduction and on Discussion. The paper approaches only YT channels, but further discussions will refer also to other social media channels.

There are countless research articles related to YT, the topic of this paper is unclear as a theme throughout the manuscript so originality is difficult to assess.

We have not found any research using Topsis method and semantical analysis for describing the channel on the scientific literature.

The Lit Rev requires extensive strengthening to support the hypothesis development, th foundation for the RQs is not clear, the background of the topic, existing related literature on the topic is sparse.

We have added literature review. And also, we have added the following explanation: through the use of semantical analysis, we have transformed information into data that may be statistically analyzed, and through the use Topsis method, we have measured data regarding subscribers, views and uploads.

The development of a the RQ based on existing literature is difficult to ascertain.

We have added that there is no statistical analysis based on data. The descriptions were transformed from information to data, thus covering the knowledge gap.

The Hypotheses seem to come out of nowhere, without referencing supporting prior related research.

We have added literature review and explanations regarding prior related research.

The connection between the two Hypotheses seems to be tenuous. Almost unrelated.

SB is not defined the first time it is used.

A successful online business has to have both elements: description and subscribers, views and uploads. We also have defined now SB used for the first time.

“Statistics are freely available to anyone using our website or smartphone apps”. This sentence is concerning, it is in the first person and appears to have come from the SB website?

The sentence was modified and better explained.

The selection and application of the methodology and the analytical tools lacks justification.

The explanation for the using of SB, as instrument for the research, was added - The reason for using Social Blade as instrument for this research is that SB is an American platform founded in 2008, widely recognized for statistical studies, a new and modern instrument used mostly by social media entrepreneurs that start a new business on YouTube. It offers information about subscribers, too.

As for the methodology, we added explanations – we consider that using Topsis method for comparing the channels with an ideal, is a very proper instrument to study their hierarchy. It is a method of compensatory aggregation that compares a set of alternatives by identifying weights for each criterion, normalizing scores for each criterion and calculating the geometric distance between each alternative and the ideal alternative, which is the best score in each criterion. 

We have added that: The instruments – multidimensional data analysis and semantical analysis - are very modern and extremely efficient for analyzing large data volumes. 

The actual analysis of the data seems to be methodologically sound.

The originality of the paper consists exactly on data analyzed and conclusions obtained.

The results lack a clear connection to the H.

We have added on Discussion section the connection between the results and the hypothesis and the fact that the hypothesis are confirmed.

There are some interesting findings in this study, but the connections to the earlier sections of the paper is lacking.

We have added on Introduction and methodology section that the topic lack research on prior papers. Also, because we added literature review and explanations regarding the hypothesis, methodology and instruments, the connection seems to be more defined now.

The Discussion does not appear to connect to the earlier sections of the paper, and lacks a clear explanation of the theoretical contribution. The practical contribution is brief and not clearly connected to the premise of the paper.

We have added theoretical contributions and we have also added explanations regarding the gap of knowledge (there is no other research using these methods and instruments) and the way the research solves the gap. Also, we added that that the paper is only a starting point, we want to continue the analyses also for other social media channels.

The References are relevant and sufficiently recent, but do not provide sufficient foundation for the paper (see above).

We have added some more literature review.

Best regards,

Dana and Radu

Reviewer 2 Report

Dear author, please consider the following suggestions.

The hypothesis should be developed in the literature review section with strong arguments

Furthermore, the author (s) should adopt Kendall’s coefficient of concordance to analyze factors, moreover, rank the factors based on the severity index.

The conclusion and discussion should not be together, please discuss them separately

The author should discuss the study implications in terms of practical, theoretical, and methodological.

Author Response

Dear Reviewer 2,

Thank you very much for your reviews! Your observations are very valuable for us, we think that they enriched our research and that the insights and the results will be even better received, understood and even more useful for scientist and large public as well, thus, we have been reading and solving each one, very carefully. 

In the following part, we tried to answer your requirements.

The hypothesis should be developed in the literature review section with strong arguments

We have added Content about literature review in order to build a foundation for hypothesis development. The strength of the paper and the need of this study are that there is no other research on this particular topic – how to choose a domain and how to describe the channel when starting a YT business as a social media entrepreneur.

The explanation for the using of SB, as instrument for the research, was added - The reason for using Social Blade as instrument for this research is that SB is an American platform founded in 2008, widely recognized for statistical studies, a new and modern instrument used mostly by social media entrepreneurs that start a new business on YouTube. It offers information about subscribers, too.

As for the methodology, we added explanations – we consider that using Topsis method for comparing the channels with an ideal, is a very proper instrument to study their hierarchy. It is a method of compensatory aggregation that compares a set of alternatives by identifying weights for each criterion, normalizing scores for each criterion and calculating the geometric distance between each alternative and the ideal alternative, which is the best score in each criterion. 

Furthermore, the author (s) should adopt Kendall’s coefficient of concordance to analyze factors, moreover, rank the factors based on the severity index.

The paper was not centered on the importance of the factors and their concordance, but it is our focus for future research. Thus, the research will be enriched with the use of Kendall’s coefficient of concordance.

The conclusion and discussion should not be together, please discuss them separately

We have made the change, and now, the conclusions and the discussions are separately.

Best regards,

Dana and Radu

Reviewer 3 Report

The title "YouTube Channels, Subscribers, Uploads and Views: An Analysis of The First 1700 Channels From July 2022"   is quite broad. It should provide some reflection on the field of study/discipline with keywords. This will help to set the reader's expectations. 

The abstract is quite short, with no description of the study's issue and objective. The significance of the research should be reflected in the abstract's conclusion.

The justification for using SocialBalde.com's data can be further discussed in the methodology. Hypothesis statement needs amendament. Discussions must be able to answer the research questions.

Author Response

Dear Reviewer 3,

Thank you very much for your reviews! Your observations are very valuable for us, we think that they enriched our research and that the insights and the results will be even better received, understood and even more useful for scientist and large public as well, thus, we have been reading and solving each one, very carefully. 

In the following part, we tried to answer your requirements.

The title "YouTube Channels, Subscribers, Uploads and Views: An Analysis of The First 1700 Channels From July 2022"   is quite broad. It should provide some reflection on the field of study/discipline with keywords. This will help to set the reader's expectations. 

The title was improved: "YouTube Channels, Subscribers, Uploads and Views: A Multidimensional Analysis of The First 1700 Channels From July 2022". We use multidimensional analysis syntagma because we have analyzed through may perspectives 1700 YouTube channels.

The abstract is quite short, with no description of the study's issue and objective. The significance of the research should be reflected in the abstract's conclusion.

We have added, on Introduction, explanations regarding the purpose, the objective and the significance of the paper.

The justification for using SocialBalde.com's data can be further discussed in the methodology. Hypothesis statement needs amendament. Discussions must be able to answer the research questions.

We have added explanations regarding the use of Social Blade, Topsis method and also, on discussion, the fact that hypothesis are confirmed.

The explanation for the using of SB, as instrument for the research, was added - The reason for using Social Blade as instrument for this research is that SB is an American platform founded in 2008, widely recognized for statistical studies, a new and modern instrument used mostly by social media entrepreneurs that start a new business on YouTube. It offers information about subscribers, too.

As for the methodology, we added explanations – we consider that using Topsis method for comparing the channels with an ideal, is a very proper instrument to study their hierarchy. It is a method of compensatory aggregation that compares a set of alternatives by identifying weights for each criterion, normalizing scores for each criterion and calculating the geometric distance between each alternative and the ideal alternative, which is the best score in each criterion. 

We have added that: The instruments – multidimensional data analysis and semantical analysis - are very modern and extremely efficient for analyzing large data volumes. 

Best regards,

Dana and Radu

Reviewer 4 Report

This study presents a new finding. However, I have a few comments below that will enrich the work.

Topic: YouTube Channels, Subscribers, Uploads and Views: An Analysis of The First 1700 Channels From July 2022

Abstract

The abstract is well written. It captures all the very important aspects of the study. I suggest that a brief statement of the advantages of the proposed method should be stated in the abstract.

Introduction

 This section is well written. However, there are a few typos that need to be corrected. Carefully read through this section to effect the corrections.

 Methodology

Please standardize English expressions, and adjust page layout, paragraph indentation, and reference format.

Authors should check the article for typos and grammatical errors. In general, the typeset equations should be regarded as parts of a sentence and treated accordingly with the appropriate grammatical convention and punctuation. More editing for writing is needed. At the end of all equations must be put "COMMA" or "POINT" according to the typing rules. The author must use the latex journal form and use the correct latex relations, in particular, it seems that mathematical relationships are not written with latex.

Rows (127) to (139)  and (167) to (185) contain plagiarism. Please rewrite them to avoid plagiarism.

Results 

This section could be better written and explain more about the tables and figures.

Row (258) " Table 1" instead of " Figure 05", seems to be a table, also, correct the table at the beginning of page 8.

Conclusion   

Authors should include limitations of the method in a subsection in the conclusion section. However, there are a few typos which need to be corrected. Carefully read through this section to effect the corrections.

Author Response

Dear Reviewer 4,

Thank you very much for your reviews! Your observations are very valuable for us, we think that they enriched our research and that the insights and the results will be even better received, understood and even more useful for scientist and large public as well, thus, we have been reading and solving each one, very carefully. 

In the following part, we tried to answer your requirements.

This study presents a new finding. However, I have a few comments below that will enrich the work.

This was exactly the scope, to cover a gap on knowledge regarding the subject.

Topic: YouTube Channels, Subscribers, Uploads and Views: An Analysis of The First 1700 Channels From July 2022

Abstract

 The abstract is well written. It captures all the very important aspects of the study. I suggest that a brief statement of the advantages of the proposed method should be stated in the abstract.

We have added, on Abstract, that there is no other study using the multidimensional analysis and Topsis method, also the objective. The significance of the paper covers the gap of knowledge.

Introduction

 This section is well written. However, there are a few typos that need to be corrected. Carefully read through this section to effect the corrections.

Thank you for your remarques, we have made changes for the paer to be more cursive!

Methodology

Please standardize English expressions, and adjust page layout, paragraph indentation, and reference format.

 Authors should check the article for typos and grammatical errors. In general, the typeset equations should be regarded as parts of a sentence and treated accordingly with the appropriate grammatical convention and punctuation. More editing for writing is needed. At the end of all equations must be put "COMMA" or "POINT" according to the typing rules. The author must use the latex journal form and use the correct latex relations, in particular, it seems that mathematical relationships are not written with latex.

We have modified and some terms and categories are used as they are, for a long time.

Rows (127) to (139)  and (167) to (185) contain plagiarism. Please rewrite them to avoid plagiarism.

The rows were rewritten.

Results 

This section could be better written and explain more about the tables and figures.

Row (258) " Table 1" instead of " Figure 05", seems to be a table, also, correct the table at the beginning of page 8.

The modification is done and also more explanations added.

There is a strong correlation between the number of subscribers and the views (0.920) and almost no correlation between the number of uplods and the number of views (0.095)

Data are real, relevant and very up to date, thus, the results may be used in case of decisions regarding social media entrepreneurs, for new sustainable managerial behaviours.

Conclusion   

 Authors should include limitations of the method in a subsection in the conclusion section. However, there are a few typos which need to be corrected. Carefully read through this section to effect the corrections.

The study was conducted only for YouTube as a source for data, if we have used different sources, the results would have been different, and also we have the research is grounded on YouTube channel description in English – if we have used other descriptions, for other languages, the ensemble images would have been more notable.

Best regards,

Dana and Radu

Round 2

Reviewer 1 Report

Your revisions have addressed my comments. 

Nice paper. 

Reviewer 2 Report

Accepted please 

Reviewer 3 Report

Satisfied with the improvement & revision made by the authors.

Reviewer 4 Report

 Accept in present form